# Experiences and challenges of parents caring for children with attention-deficit hyperactivity disorder: A qualitative study in Dar es salaam, Tanzania

**Charles Daud Ching'oma**[1☉], **Dickson Ally Mkoka**[2‡], **Joel Seme Ambikile**[2☉]*, **Masunga Kidula Iseselo**[2‡]

1 Lugalo General Military Hospital, Dar es Salaam, Tanzania, 2 Clinical Nursing Department, School of Nursing, Muhimbili University of Health and Allied Sciences, Dar es Salaam, Tanzania

☉ These authors contributed equally to this work.
‡ DAM and MKI also contributed equally to this work.
* joelambikile@yahoo.com

## Abstract

### Background

Attention-deficit hyperactivity disorder (ADHD) is the most common neurobehavioral childhood disorder. Children with ADHD are difficult to handle due to the symptoms causing great impairments such as inattention, hyperactivity compared to other childhood mental disorders. Having a child with ADHD is a stressful situation as it impacts the whole family. However, little is known about the experiences and challenges of parents caring for children with ADHD in low and middle-income countries such as Tanzania. Thus, this study explored the experiences and challenges of parents caring for children with ADHD in Dar es Salaam, Tanzania.

### Methods

We conducted a qualitative descriptive study involving 16 parents of children with ADHD at Muhimbili National Hospital (MNH). We used the purposive sampling technique to obtain the participants. In-depth interviews, using a semi-structured interview guide, were used to collect data. Audio-recorded data were transcribed, translated, and analysed using qualitative content analysis.

### Results

Parents experienced difficulties in handling the children whose level of functioning was impaired due to abnormal and disruptive behaviour such as not being able to follow parental instructions. Psychological problems were also experienced due to caring demands exacerbated by lack of support and stigma from the community. Moreover, there were disruptions in family functioning and social interactions among family members due to the children's

**Data Availability Statement:** All relevant data are within the paper and its Supporting Information files.

**Funding:** The author(s) received no specific funding for this work.

**Competing interests:** The authors have declared that no competing interests exist.

behaviour. Lastly, too much time and family resources spent to fulfil the needs of these children culminated into disruption in economic activities that negatively affected everyday life.

## Conclusion

Parents struggle to meet and cope with care demands posed by children with ADHD. The disruptive nature of ADHD symptoms presents a unique caring challenge different from those experienced with other childhood mental illnesses. To address these challenges, a collaborative approach among key stakeholders such as the government, health care professionals, and non-governmental organizations, is needed.

## Introduction

Attention deficit hyperactivity disorder (ADHD) is the most common neurobehavioral childhood disorder associated with inattention, hyperactivity, and impulsivity which surface between 3 and 7 years (American Psychiatric Association, 2013). It is approximated that 2.0% to 16.0% of children are affected by ADHD globally [1, 2]. In the USA and Iran, the prevalence of ADHD ranges between 2–20%, while in Congo and Nigeria, it is 6% and 8% respectively [3]. Despite all the previous studies conducted on ADHD, the challenges from this disorder are still increasing [4]. Prevalence of ADHD varies worldwide due to demographic, cultural, socioeconomic, and the criteria used for diagnosis. The disorder occurs in all socioeconomic groups, especially in low income countries, where evidence is still scarce [5].

ADHD has deleterious effects among children, both at school and home settings. It results in restlessness, impulsive acts, and lack of focus, which may impair the child's performance [6]. These symptoms are challenging to manage particularly by parents with limited skills and understanding of the child's outcomes [7, 8]. In this context, parents are the first-line caregivers, who often experience a heavy burden of care associated with the disorder. Previous studies in Tanzania and Palestine reported that the burden of care significantly affects parents' physical and mental health [9, 10].

Most parents caring for children with ADHD experience similar effects, though some variations may occur due to social-economic factors and geographical locations [10]. For instance, in France parents experienced intense emotions and physical exhaustion [11]. Similarly, in South Africa parents experienced difficulties such as negative emotions, economic problems, inadequate social support, stigma, and extra caregiving responsibilities [6]. The burden of care experienced is substantial, leading to strained family relationships and stigma, coupled with minimal support [12]. Additionally, parents experience disruption in family functioning such as spouse misunderstanding, unnecessary divorces, and financial constraints [13, 14].

Previous studies have highlighted the role of parenting skills training in improving family relationship [15]. Research has also demonstrated that parents who have continuous contact with mental health professionals are more likely to recover from the crisis and cope with challenges of caring for mentally ill children [9]. To date, Tanzania lacks established statistics on families affected by mental illnesses [14]. Thus, this study explored the experiences of parents of children with ADHD at Muhimbili National Hospital, Dar es Salaam, Tanzania.

## Materials and methods

### Study design

We employed an explorative study design using the phenomenology qualitative approach. The choice of this design was dictated by the nature of our study as there was limited information

on the challenges and experiences of parents caring for children with ADHD. Thus, we wanted to familiarize with the basic concerns and get a well-grounded picture of the caring situation for children with ADHD in the local context. The design is also used when the problem has not been clearly defined [16].

## Study context

The study was conducted at Muhimbili National Hospital (MNH), in Dar-es-Salaam, Tanzania. MNH provides the most advanced specialized health services, where patients with complicated health problems are referred to from all over the country. Mental health services are provided in two major forms, i.e. on inpatient and outpatient basis, with adult and child and adolescent services provided separately. The clinics are conducted three days per week; on Monday, Wednesday, and Friday. The clinics also receive new children with mental illnesses from all over the country. Serious patients who require long-term hospitalization are referred to Mirembe National Psychiatric Hospital in the Dodoma region. The child and adolescent clinic serves approximately 50–55 patients every week according to the 2020 clinic records. Psychiatrists, mental health nurses, and social workers are the forefront professionals providing care to mentally ill children.

## Study participants

Participants recruited in this study were parents of children with ADHD. A parents in this study referred to a biological mother, father, or any other person with caregiving responsibility to the child. An inclusion criterion was a parent who had stayed with the child for more than six months and directly involved in the caring process. The six months period is considered an adequate time for having reasonable caregiving experience to the child with ADHD [17].

## Selection of participants

We used purposive sampling technique to obtain study participants. This technique enabled us to get participants with adequate experience and information to answer our research questions. Potential participants were obtained from the files of children with ADHD at the psychiatric and mental health unit. Those who met the criteria were contacted physically and through phones and asked to participate in the study. Therefore, 25 eligible participants agreed to participate and provided their phone contacts for future communication and appointment to attend the interview. However, 8 did not show up for various reasons and 2 could not be reached through their mobile phones until the end of data collection.

## Data collection methods and tools

In-depth interview was used as a data collection method. A semi-structured interview guide was used to collect the information from participants. The interview guide was based on a recent literature review as well as the researchers' clinical experience in caring for children with ADHD. The tool was pre-tested and after revision the final interview guide had 5 main questions as shown in the Table 1 below:

The interview guide questions were followed by specific probes to get a deeper understanding of the participants' experiences.

The data was collected between 4th June and 10th July 2020. Appointments were made two days before the day of the interview by a research assistant who was a nurse with a background in qualitative studies. On the day of the interview, participants were met at the clinic at the agreed time. All interviews were conducted in Kiswahili, the common language spoken by

**Table 1. Interview guide.**

| |
|---|
| 1. How do you handle the disruptive behaviours of your child? |
| 2. Can you explain any social or emotional support you received from the society? |
| 3. What are the social and family dysfunctions you experienced after your child diagnosed with ADHS? |
| 4. Can you explain the relationship of the child with ADHD with other children without ADHD in the family? |
| 5. How this child with ADHD has affected your relationship with your spouse? |

participants. The interviews were carried out at the Child and Adolescent Psychiatric clinic in a room that was available and temporarily prepared and used for this purpose. The room provided privacy, was quiet, and had good light for proper observation of nonverbal cues. Before the interviews, participants were briefed about the aim of the study and their rights to participate, including the right to quit from the study. The interviews were recorded using a digital recorder and continued until information saturation was attained at 16. The mean duration of the interviews was 45 minutes.

## Data analysis

We conducted manual analysis which commenced soon after the first interview. The interviews were transcribed verbatim and translated into English. Authors read the transcripts and compared them against the original text and against the audio records to ensure coherence in content. To maintain validity, we consulted the original transcripts regularly to ensure interpretations were grounded in the data. Kiswahili and English transcripts were kept together for easy crosschecking.

The content analysis approach, a research tool used to determine the presence of certain words or concepts within texts or sets of texts, guided the data analysis process [18]. This is the widely used research technique to analyse qualitative data which involves 4 stages, i.e. decontextualisation (identifying meaning units and creating code list), recontextualisation (including "content" and excluding "dross"), categorisation (identifying homogeneous groups), and compilation (drawing realistic conclusions) [19]. Therefore, we first read and re-read the transcripts several times to gain a general impression of the contents and familiarize with the data. Then we identified parts of transcripts that corresponded to the study objectives as meaning units of the transcript. The identified meaning units were then condensed closely adhering to the text. Codes that carried interpretation of underlying meanings were extracted from each condensed meaning unit. Based on the similarities or differences, the extracted codes were grouped into sub-categories and categories reflecting the core meaning of the text (the manifest content of the text). We then reviewed, defined, and then discussed intensively for their relevance with research questions and discussed the analysis outcome and reached a consensus. We kept interviews as the point of reference when deeper understanding was required concerning the meaning units, codes, and sub-categories. We present findings verbatim with quotes from participants as shown in Table 2.

## Ethical approval and consent to participate

Ethical clearance was obtained from the Institutional Review Board of Muhimbili University of Health and Allied Sciences with IRB No. MUHAS-REC-04-2020-26. Permission to conduct interviews was sought from the management of Muhimbili National Hospital. We explained the aim, procedure, and benefits of the study to the participants. Participants were informed of their right to participate or withdraw to ensure openness of the study and cooperation. Participants were ensured that all information would be treated with high confidentiality and remain

**Table 2. Examples of formation of categories during the analysis.**

| Meaning Unit (MU) | Condensed MU | Codes | Sub-Categories | Categories |
|---|---|---|---|---|
| "Information about this disorder and how to parent the child I had no experience. Maybe when we come to these clinics that we have started, is where we meet with a panel of doctors" | Information about ADHD and parenting skills are scarce, only obtained when contacting doctors | Lack of skills to parent the child Only one source of information—clinic Advice is only when visiting doctors | Insufficient information about ADHD Insufficient parenting skills | Need for knowledge of ADHD and child handling skills. |
| ". . ..I rush back to help my wife because there was no house girl to help to stay with the child!! They cannot tolerate that stubbornness. My wife lost her job, and she had to ask for resignation.." | Parents have to give up some important economic tasks for the sake of their children | More time needed to take care of the sick child Intolerance of child behaviours by helpers Giving up some economic activities | Increased burden of care due to intolerable behaviours The economic burden on the parents | Disruption of economic activities |
| ". . .Yah, I have to feel that way (pretending okay) even if I feel bad, but the child is already mine I cannot leave for somebody else. I have to accept it." | Parents feel sorry and overwhelmed about their children with ADHD | Parents feel bad about themselves Lost hope of support from somebody else | Emotionally overwhelming Physical exhaustion on caring for the child | Emotional and physical exhaustion |

within the intended aim of the study and that their names would not be used in any record. Written informed consent to participate and record the interview was obtained from each participant.

# Findings

## Sociodemographic characteristics

Table 3 shows sociodemographic characteristics of participants. Of the 16 participants, 9 were aged 40 years and above, 12 were biological mothers, 8 had primary education, 10 were

**Table 3. Social demographic characteristics.**

| Characteristic | Type | Frequency/Number |
|---|---|---|
| Age of parents | Between 30 and 39 | 5 |
| | 40 years and above | 9 |
| Age of children with ADHD | 6 to 10 years | 12 |
| | 11 to 15 years | 4 |
| Sex of children with ADHD | Males | 13 |
| | Females | 3 |
| Type of parents | Biological fathers | 4 |
| | Biological mothers | 12 |
| Level of Education | Primary education,. | 8 |
| | Secondary education, | 5 |
| | College education | 3 |
| Marital Status | Single | 2 |
| | Married | 10 |
| | Divorced | 4 |
| Employment Status | Employed | 5 |
| | Self-employed | 10 |
| | Not employed | 1 |
| Residence | Urban | 6 |
| | Rural | 10 |

married, 10 were self-employed, and 10 were coming from a rural area. Most of their children with ADHD were males (13) and aged 6 to 10 years.

## Themes

During analysis four main themes emerged from the data. These were difficulty in handling a child's abnormal behaviour, psychological problems due to caring demands, disruption of family functioning and social stability, and disruption of economic activities within the family. Each theme had categories as shown in Table 4.

**Challenges in handling child's abnormal behaviour.** *Child's safety concern*. Participants in this study reported having hard times caring for children with ADHD. They reported that children begun to display the abnormal behaviour at an early age, raising serious concerns in their daily roles and affecting their performance. They further expressed that, as children grew up, their abnormal behaviour resulted in more significant challenges at home, particularly on their safety. Being easily distracted, trouble with listening, constantly moving here and there, and being exposed to injuries created a big burden to the whole family as expressed by a female parent:

> *"Doctor this child. . .I cannot go to public places or use Daladala (commuter bus) together with him; he grabs items from people and sometimes hits people who try to warn him with stones. He should remain locked inside a room for the whole day". (Female, 45 years old,)*

Living with a child with ADHD was found to be very demanding, requiring some participants to modify accommodation to meet their child's needs. Some of the participants reported using harsh physical punishment in trying to rectify their children's disruptive behaviour without any success as stated by a male parent:

> *"I always use sticks beating him but this child is not listening and does not seem to change. My wife is struggling and puts a lot of effort to change and make him like other children. . .She does a lot"* (*Male, 40 years old*).

*Parental reaction to child' disruptive behaviour*. Participants reported that it was difficult for the children to follow daily routines, such as abiding by the daily rules. Behaviours like beating other children, fighting with a parent, and destroying property troubled parents who confessed to have insufficient skills to discipline their children. Some parents were puzzled and didn't know what to do, others tried to channel the child's energy by giving them activities to do, while still others took inappropriate disciplinary measures such as reprimanding and use of corporal punishment when children overtly misbehaved. Male parents tended to be more

**Table 4. Themes and categories.**

| Themes | | Categories |
|---|---|---|
| Challenges in handling child's abnormal behaviour | 1 | Child's Safety concern |
| | 2 | Parental reaction to child' disruptive behaviour |
| Psychological problems associated with caring demands | 1 | Lack of emotional support |
| | 2 | Social discrimination |
| Family and social dysfunctions | 1 | Disrupted family process |
| | 2 | Disrupted neighbourhood relationship |
| Disruption of economic activities within the family | 1 | Lack of household manpower |
| | 2 | Lack of financial support |

violent and aggressive towards the child and sometimes extended their anger towards the spouse as stated by a female participant:

"...*One day my husband was beating the child too terrible and I was shocked (...) when I tried to stop him, he turned on to me. Always beating the child but it does not help*" (Female,49 years old).

**Psychological problems associated with caring demands.** *Lack of emotional support.* Some parents expressed feelings of being exhausted emotionally. They verbalized feelings of depression and sadness due to care demands. They reported that caring for children with ADHD was stressful, requiring continuous reassurance and emotional support from teachers and psychologists. Furthermore, participants reported that they had been interacting with other parents and hoped for acceptance and inclusion in the society. Some reported that their children also needed emotional support from grownups and friends, but they had no friends. Generally, participants demonstrated to be emotionally overwhelmed with caring demands which were said to be stressful and difficult to cope with.

"...*Yah, I have to feel that way (pretending to be okay) even if I feel bad, but the child is already mine I cannot leave (him to someone else. I have to accept the way it is and wait for God to help in the care...*" (Female,42 years old).

The disruptive behaviour exhibited by children both at school and home was reported to cause emotional difficulties to parents. They expressed that, they often turned to schoolteachers and health professionals for guidance. Self-blaming and being isolated by the community were major concerns experienced by the parents as stated below:

"*Sometimes I feel so alone and would like to chat with other parents about how they handle it all, it can all be so hard. I think that would also be helpful to hear [...] from the other parents as well...*". (Female, 38 years old)

*Social discrimination.* Participants expressed that they felt abandoned and experienced discrimination from the community for having children with abnormal behaviour and felt they could benefit from peer support and other social networks if they were connected with the world. It was noted that caring for children with ADHD brought a sense of unworthiness and social isolation which had a negative impact on their health and well-being. Despite having family support, they reported feelings of increased isolation, which contributed to distress as expressed by a female participant:

"*I don't feel worth living, I am socially isolated and lack support from my relatives, friends, and health professionals. I think there is no use to tell others to get help or advice because they don't understand me*". (Female,30 years old)

Social discrimination occurred even in the political affairs as they reported to be discriminated against and isolated when contesting for leadership positions. Lack of understanding of ADHD in the society underpinned political problems participants experienced as stated below:

"*Last year there was an election to our savings and credit cooperative society, I decided to take part to be chosen as one of the committee member, but they did not accept my name as some*

*of the members thought that I am also sick and my child has inherited ADHD from me. . .thus I am not fit to be a leader". (Female,38years old)*

**Family and social dysfunctions.**    *Disrupted of family process.* Participants said that life in families with a child with ADHD was influenced by the child's behaviour, parental role, family functioning, and support from the social network. They insisted that family functioning is crucial in managing life such as problem-solving and social networking roles. Female participants reported that fathers always blamed them for always favouring and being too fair to the children. This stressful situation created conflicts in families and led to unnecessary divorces as stated by a female participant:

*"I'm a single mother with two children. . . one is okay, but this one (sick child) only God knows. . .their father is enjoying life there. . . we divorced two years ago and he does not care or even visit us." (Female,37years old)*

During a typical day, extra time was required by participants to make sure children got their needs. They reported to have lower self-confidence and less warmth and involvement with their children and used corporal punishment more than other forms of interventions, putting children at risk of abuse.

*"Disruptive behaviours of my child caused him to have academic underperformance, disciplinary issues at home and school. This affects all areas of family life such as relationships in the family and relatives". (Female,30 years old)*

*Disrupted neighbourhood relationship.* Parents reported that their relationship with neighbours was not good as they were blamed for their children's behaviour. Their children were often beaten when they entered neighbours' houses and the family sometimes expelled from the house by the landlords. Parents were always worried about how the child would behave if they got visitors or went to public places and their social life was affected as narrated by a male participant:

*". . .When we get visitors at home; they will not stay long . . .I remember one day I attended a party at my neighbour's house, what he (the child) did was horrible . . .beating other children. . . taking off clothes, threatening to beat other children around. . .. I decided to leave immediately" (Male, 32 years old)*

Lack of tolerance among neighbours towards child's behaviour was also reported by some parents. On many occasions, they had to pay for or buy the neighbours' items to compensate for what the child had destroyed. The neighbours were not kind enough to forgive the child and maintain good terms with the child's family, as expressed by a male participant:

*"Last year we had a case at the police station with one of the neighbours after a long term conflict with them, they demanded that we should lock our child inside after breaking their 40 inch flat-screen television. It's too expensive for us to pay but they still insist that we should pay back their money or buy the flat-screen television." (Male, 40 years old)*

**Disrupted family economic activities.**    *Lack of household manpower.* Almost all parents reported going through the economic crisis and voiced out for social-economic support. It was noted that the care demands of children with ADHD interfered with the family's economic

activities, leaving the family unstable economically. One parent revealed that his wife had to stop working to take care of the child after their house girl left due to the child's disruptive behaviours:

"…. I rush back to help my wife because there was no house girl to stay with the child. All house girls ran away despite promising to pay them a good amount of money!! They could not tolerate the child's stubbornness. My wife was expelled from her job." (Male,47 years old).

In addition, parents reported having limited social network affiliations and interpersonal relationships at work place. It was very difficult to get employed because of their poor social network which deprived the family of meeting essential needs as a result of having a sick child. They also had no time to engage in other economic activities as they were busy supervising the child most of the time.

*Lack of financial support.* Participants reported that their families were poor because they had no support even from any of their relatives. Their family resources were consumed by treatment costs which necessitated even selling a house or the land they had. They also sacrificed the needs of other children to meet the needs of the child with ADHD as expressed by a female participant:

*Our child (a healthy one) is being expelled from school after we failed to pay school fees, my husband is trying a lot but we have remained poor since then …what we get all goes to manage our (sick) child" (Female, 54 years old).*

Participants expressed the need for more support in areas such as education and health services. Schools for children with ADHD were expensive and parents could not afford sending them there, hence many children stayed at home. They also reported having difficulties accessing healthcare services for their children since it was costly as stated by a female participant:

*"It is very expensive . . . the medication prices are horrible . . .. My biggest worry is that he will be taking these medications for life with no health insurance" . . .. (Female, 38 years old).*

Some participants were blamed by doctors when they failed to get medications for their children. They reported unsupportive attitudes from mental health professionals when they expressed their experiences and challenges, including problems with affording transport fair to the hospital.

## Discussion

This study aimed at exploring caregiving experiences of parents of children with ADHD at Muhimbili National Hospital, Dar es salaam, Tanzania. Most participants were females and the findings reveal that they experienced difficulties in handling children's abnormal behaviour and emotional and physical exhaustion. Participants also reported disruption of the family process and social interaction. In addition, disruption of economic activities was another important reported finding in the current study.

Most of the participants in this study were females. This was not caused by the purposive sampling technique we used but the clinic records showed that females were the main caretakers of the children. This is not a new finding since many previous studies within and outside Tanzania have shown that, unlike their male counterparts, female parents and guardians bear a greater burden of care, not only to children, but also to adults and the elderly with mental

illness [9, 20, 21]. This may be explained by the influence of culture and ethnicity which are known to have a seminal influence on caregiving [22].

As reported in our study, difficulties in handling the child's abnormal behaviour is a frustrating situation among many caregivers. The fact that children do not follow instructions when instructed, or never concentrate, may be described by the psychopathology of ADHD in children. ADHD symptoms affect the child's behaviours resulting in significant challenges at home and schools [23]. Difficult handling was fuelled by children's academic underachievement, disruptive conduct, and having poor peer relationship. This finding is consistent with that reported by Mohr-Jensen and colleagues whereby parents expressed their concerns about their children's poor peer relationship [24]. Moreover, participants faced these difficulties because they were not cognizant of the clinical symptoms of ADHD. Similar experiences of handling children with abnormal behaviours have been reported in France [11], although parents were more informed and aware of ADHD than those in our study. The difficulties in handling abnormal behaviour reported in our study prevented parents from planning for children's basic needs. This important area needs further investigation.

The emotional and physical exhaustion experienced by the parents was driven by children's abnormal and disruptive behaviour. This situation has adverse consequences on the affairs of the family to fulfil parental roles. This finding is consistent with the previous study [9] which revealed the emotional and physical effects when caring for children with mental illness, including ADHD. Participants experienced a lack of sleep due to monitoring their children night long. A prolonged lack of sleep affects physical and mental well-being [25]. Additionally, parental stress might have been exacerbated by the co-occurrence of conduct disorders in children with ADHD [26, 27], especially when the child is not using medications regularly. The shortage of medications caused stress to the parents and children. Lack of medication may be due to problems with availability in the government stores as found in the previous study [14]. In the context of emotional and physical exhaustion experienced by parents, continued psychological support from health professionals is necessary. Research shows that when participants are well supported and collaborate with health professionals their care burden may be lessened [28].

The disruption of family process and social stability experienced by the parents created substantial impairments in social and peer functioning in the families. Parents experienced disturbances in their normal routines as a result of having an ADHD child at home. In such a situation, normal family functioning is affected by the child's daily behaviour [29]. This finding is similar to the experiences of parents reported by the study conducted in Norway whereby caregiving conflicts between the spouses occurred frequently [30]. This argument is supported by a systematic review, which showed that female caregivers experience more caregiving burden than males [31]. This discrepancy in caregiving burdens causes family conflicts and misunderstandings among the spouses as reported in our findings. The finding is also consistent with the study conducted in Nairobi, Kenya [32], where mothers were mostly blamed due to the bad conduct of their children. Such similarity may be explained by the fact that Kenya and Tanzania have similar social-cultural practices. The fact that parents were more likely to divorce due to the caregiving burden needs further investigation as marital conflict and divorce may be contributed by other factors [33]. Family disruption was also revealed in the previous study [10] whereby social disintegration was reported by many families. More research is needed to reveal how better professional support can be provided to promote family functioning for parents caring for children with DHD in similar settings.

The disruption of economic activities occurred due to the parents experiencing inadequate support from the government and community. Caregiving remained as their role while having other responsibilities in the family. This is consistent with the previous study in the United

Kingdom [34]. The financial constraints experienced by the participants in this study may be contributed by participants staying at home most of the time to look for sick children. Also, the tendency of many house girls (housemaids) leaving their job as reported in this study may be attributed to the intolerable behaviours of children with ADHD. This is because, caregiving for children with ADHD is considered a "24 hours, 7 days a week task" which involves constant monitoring of the children to prevent injurious behaviours towards self or others. This experience is also reported elsewhere [9]. There is no specific policy to deal with children with ADHD, which makes parents lack support from the community and government. As reported in our study, traveling long distances to attend multiple clinic appointments contributes to disrupted economic activities among participants. Sometimes, they need to sacrifice the needs of other children in the family to meet the needs of one child as reported also by Oruche et al. who highlighted that family resources are used to care mostly for the sick child [35]. This is also corroborated by a previous study [36] where parents cried for financial constraints with no or minimal community support. This was due to the amount of money earned from the limited working time that was all used to care for the ADHD child such as purchasing medications.

The strength of this study is that we used the purposive sampling technique to recruit participants which enabled us to get in-depth and detailed information about the experiences and challenges parents of children with ADHD go through. The limitation is that we did not explore variations of experiences and challenges among parents which could be influenced by sociocultural factors, including the sex of children which in our study was predominantly males. The scope of this study precluded a detailed consideration of cultural aspects of caring and we suggest further research to explore the needs of diverse cultures, social classes, and communities.

## Conclusion

Parents living with a child with ADHD experience various challenges as they struggle to cope with the burden of care. Caring for children with ADHD is different when compared to other chronic illnesses due to the nature of ADHD symptoms. Parents experience multiple social, economic, and psychological challenges due to care demands. Support from the government and other stakeholders is important for the well-being of both parents and their sick children. Addressing these problem requires use of a collaborative approach among the government, health care providers, and other stakeholders. Further studies including quantitative research to investigate issues such as parents' stress levels and depressive symptoms are needed.

## Supporting information

**S1 Data.**
(DOCX)

## Acknowledgments

Much appreciations go to Dr. Beatrice Mwilike from MUHAS school of Nursing and Leticia Lungulungu for their important contributions such as editing the manuscript. A lot of thanks go to Madam Shamila from MNH who helped in data collection and conducting interviews. Finally, thanks go to MNH where the study was allowed to be conducted.

## Author Contributions

**Conceptualization:** Charles Daud Ching'oma.

**Data curation:** Charles Daud Ching'oma.

**Formal analysis:** Charles Daud Ching'oma, Dickson Ally Mkoka, Joel Seme Ambikile.

**Investigation:** Charles Daud Ching'oma, Joel Seme Ambikile, Masunga Kidula Iseselo.

**Methodology:** Dickson Ally Mkoka, Joel Seme Ambikile.

**Project administration:** Charles Daud Ching'oma.

**Resources:** Charles Daud Ching'oma.

**Supervision:** Dickson Ally Mkoka, Joel Seme Ambikile, Masunga Kidula Iseselo.

**Writing – original draft:** Charles Daud Ching'oma, Joel Seme Ambikile, Masunga Kidula Iseselo.

**Writing – review & editing:** Dickson Ally Mkoka, Joel Seme Ambikile, Masunga Kidula Iseselo.

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
