## [Decision Letter · Decision Letter 0]

16 Jun 2022

PONE-D-22-11196Experiences and challenges of parents caring for children with attention-deficit hyperactivity disorder:  A qualitative study in Dar es salaam, TanzaniaPLOS ONE

Dear Dr. Ambikile,

Thank you for submitting your manuscript to PLOS ONE. After careful consideration, we feel that it has merit but does not fully meet PLOS ONE’s publication criteria as it currently stands. Therefore, we invite you to submit a revised version of the manuscript that addresses the points raised during the review process.

We look forward to receiving your revised manuscript.

Kind regards,

M Atiqul Haque, MBBS, MPH, PhD

Academic Editor

PLOS ONE

Journal Requirements:

Reviewers' comments:

Reviewer's Responses to Questions

**Comments to the Author**

1. Is the manuscript technically sound, and do the data support the conclusions?

Reviewer #1: Yes

Reviewer #2: Partly

2. Has the statistical analysis been performed appropriately and rigorously? 

Reviewer #1: N/A

Reviewer #2: N/A

3. Have the authors made all data underlying the findings in their manuscript fully available?

Reviewer #1: Yes

Reviewer #2: No

4. Is the manuscript presented in an intelligible fashion and written in standard English?

Reviewer #1: Yes

Reviewer #2: Yes

5. Review Comments to the Author

Reviewer #1: Minor Revision;

The Manuscript “Experiences and challenges of parents caring for children with attention-deficit hyperactivity disorder: A qualitative study in Dar es salaam, Tanzania” explore how parents struggle to meet and cope with care demands posed by children with ADHD.

It has got significance for the public health practices.

Please mention the study approach i.e., Phenomenology.

Provide the Interview guide in Tabular form.

How many participants were initially contacted and how many did not show their willingness to participate?

Where the interviews were carried out?

Instead of Study Limitations, write about methodological consideration with both strength and limitations.

Reduce quotation in the Results Section and describe it in the content of the results.

Reviewer #2: In title- the type of study (qualitative study) and place of study (Tanzania) need to be added.

Sample size is low (only 16 IDI), FGD can be incorporated.

Description of content analysis need to be added with reference

Gender based findings of the parents and their children with ADHD can be compared

Gender of the children with ADHD and gender based segregation of parental support can be added

Analytical findings can be expressed in percentages also (eg- how many/% of them faced challenges)

6. PLOS authors have the option to publish the peer review history of their article (what does this mean?). If published, this will include your full peer review and any attached files.

Reviewer #1: No

Reviewer #2: **Yes: **Abu Sayeed Md. Abdullah

---

## [Author Response · Author response to Decision Letter 0]

24 Jun 2022

ACDEMIC EDITOR'S COMMENTS:

Comment:

Authors' response:

We have ensured that our manuscript follow PLOS ONE's style requirements by reading and adhering to the templates found at https://journals.plos.org/plosone/s/file?id=wjVg/PLOSOne_formatting_sample_main_body.pdf and 

https://journals.plos.org/plosone/s/file?id=ba62/PLOSOne_formatting_sample_title_authors_affiliations.pdf. For this reason, we have also deleted the section named ‘Implication for practice’ as it is not supported by the sections required.

Comment:

Authors' response:

We appreciate the comment. We have made excerpts of the transcripts relevant to the study available by uploading them as Supporting Information files.

Comment:

Authors' response:

We appreciate the comment. We have uploaded our study’s minimal underlying data set as Supporting Information files.

Comment:

Authors' response:

We appreciate the comment. We have reviewed our reference list to ensure that it is complete and correct. We have also referred to the PLOSONE guidelines for preparing references to ensure that we meet the requirement. We have also added references number 20, 21, 22, and 23 due to information added in the process of addressing the reviewers’ comments

REVIEWERS' COMMENTS

REVIEWER # 1:

Comment:

The Manuscript “Experiences and challenges of parents caring for children with attention-deficit hyperactivity disorder: A qualitative study in Dar es salaam, Tanzania” explore how parents struggle to meet and cope with care demands posed by children with ADHD. It has got significance for the public health practices. 

Authors' response:

We appreciate the comments

Comment:

Please mention the study approach i.e., Phenomenology.

Authors' response:

We accept the comment. The study approach has been mentioned as phenomenology.

Comment:

Provide the Interview guide in Tabular form. 

Authors' response:

We accept the comment. The interview guide has been modified into a tabular form.

Comment:

How many participants were initially contacted and how many did not show their willingness to participate?

Authors' response:

Thank you for the comment. We accept it and have added a summary of this information in the methods section under subsection ‘Selection of participants’

Comment:

Where the interviews were carried out? 

Authors' response:

The interviews were carried in a room that was available at Child and Adolescent Psychiatric clinic which was temporarily prepared to be used for conducting interviews. This information was provided in the Materials and methods section under sub-section ‘Data collection methods and tools’. However, information about this venue for interviews has been further revised for more clarity.

Comment:

Instead of Study Limitations, write about methodological consideration with both strength and limitations. 

Authors' response:

We appreciate this good comment and accept it. We have removed Study Limitations section and included methodological consideration (with both strength and limitations) at the end of the discussion section.

Comment:

Reduce quotation in the Results Section and describe it in the content of the results. 

Authors' response:

We appreciate the comment provided. Quotations in the Results section have been reduced by deleting those which were not necessary to reduce bulkiness and the summary of information carried by deleted quotes have been included in the content of the results.

REVIWER # 2:

Comment:

In title- the type of study (qualitative study) and place of study (Tanzania) need to be added. 

Authors' response:

We appreciate the comment. However, our titles reads “Experiences and challenges of parents caring for children with attention-deficit hyperactivity disorder: A qualitative study in Dar es salaam, Tanzania” which already contains the type and place of the study.

Comment:

Sample size is low (only 16 IDI), FGD can be incorporated. 

Authors' response:

We appreciate this comment. However, we understand that in qualitative studies sample size is not the main issue. What is more important in qualitative studies is reaching ‘saturation’ with the information gathered i.e. when no new information is coming up as you continue to gather data. For this reason, we reached saturation at the 16th in-depth interview. Moreover, we did not use FGDs because we wanted to get more deeper and varying personal experiences of parents as they cared for children with ADHD which we thought could better be achieved through in-depth interviews.

Comment:

Description of content analysis need to be added with reference 

Authors' response:

Description of content analysis has been added including steps involved in the process. We have also added references to support the description.

Comment:

Gender based findings of the parents and their children with ADHD can be compared 

Authors' response:

We appreciate the comment. However, qualitatively, we were much more interested in describing experiences and challenges of parents. We think that gender based findings of the parents and their children with ADHD could be better compared in a quantitative study.

Comment:

Gender of the children with ADHD and gender based segregation of parental support can be added 

Authors' response:

We accept the comment. We have added age and gender of the children in table 3 (sociodemographic characteristics of participants). Gender based segregation of parental support is reflected in the sociodemographic data. However, since we did not explore this we have included it in the discussion section as a limitation to our study.

Comment:

Analytical findings can be expressed in percentages also (eg- how many/% of them faced challenges) 

Authors' response:

We appreciated this comment. All interviewed participants faced challenges one way or the other. In qualitative studies emphasis or interest is more placed on the quality (insight and understanding of phenomena through intensive collection of narrative data rather than quantity such as percentages or frequencies). We are of the opinion that this comment could be better addressed if this was a quantitative study.

---

## [Editor Report · Decision Letter 1]

20 Jul 2022

Experiences and challenges of parents caring for children with attention-deficit hyperactivity disorder:  A qualitative study in Dar es salaam, Tanzania

PONE-D-22-11196R1

Dear Dr. Ambikile,

We’re pleased to inform you that your manuscript has been judged scientifically suitable for publication and will be formally accepted for publication once it meets all outstanding technical requirements.

Kind regards,

M Atiqul Haque, MBBS, MPH, PhD

Academic Editor

PLOS ONE

Additional Editor Comments (optional):

We appreciate your responses to the reviewers. Now the manuscript is in a good shape to publish. Congratulations.
---

## [Editor Report · Acceptance letter]

25 Jul 2022

PONE-D-22-11196R1 

Experiences and challenges of parents caring for children with attention-deficit hyperactivity disorder:  A qualitative study in Dar es salaam, Tanzania 

Dear Dr. Ambikile:

I'm pleased to inform you that your manuscript has been deemed suitable for publication in PLOS ONE. Congratulations! Your manuscript is now with our production department. 

Kind regards, 

on behalf of

Mr. M Atiqul Haque 

Academic Editor

PLOS ONE